# Behaviour Discovery and Attribution for Explainable Reinforcement Learning

**Rishav Rishav**                                                   *mail.rishav9@gmail.com*
*University of Calgary, Mila*

**Somjit Nath**
*McGill University, Mila*

**Vincent Michalski**
*Université de Montréal, Mila*

**Samira Ebrahimi Kahou**
*University of Calgary, Canada CIFAR AI Chair, Mila*

**Reviewed on OpenReview:** *https://openreview.net/forum?id=JbHtpOIH9l*

## Abstract

Building trust in reinforcement learning (RL) agents requires understanding why they make certain decisions, especially in high-stakes applications like robotics, healthcare, and finance. Existing explainability methods often focus on single states or entire trajectories, either providing only local, step-wise insights or attributing decisions to coarse, episode-level summaries. Both approaches miss the recurring strategies and temporally extended patterns that actually drive agent behavior across multiple decisions. We address this gap by proposing a fully offline, reward-free framework for *behavior discovery and segmentation*, enabling the attribution of actions to meaningful and interpretable behavior segments that capture recurring patterns appearing across multiple trajectories. Our method identifies coherent behavior clusters from state-action sequences and attributes individual actions to these clusters for fine-grained, behavior-centric explanations. Evaluations on four diverse offline RL environments show that our approach discovers meaningful behaviors and outperforms trajectory-level baselines in fidelity, human preference, and cluster coherence. Our code is publicly available [1].

## 1 Introduction

Explaining the decisions of RL agents is increasingly important as these agents are deployed in high-stakes domains such as robotics, healthcare, and finance (Sutton & Barto, 2018; Arulkumaran et al., 2017; Fatemi et al., 2019). Interpretability is critical for building user trust, assessing safety, and diagnosing failure modes. While many existing explainability methods in RL focus on input features, individual states or entire episodes, they often overlook a crucial abstraction: *behavioral context*. Agent decisions are frequently driven by recurring strategies that unfold over multiple timesteps and appear repeatedly across episodes. This temporal structure, which reflects what the agent is doing rather than just where it is, remains underexplored in current interpretability frameworks.

One line of work focuses on identifying influential features within individual observations. These include saliency-based techniques (Greydanus et al., 2018), attention heatmaps, and attribution methods adapted from supervised learning, such as LIME (Ribeiro et al., 2016) and SHAP (Lundberg & Lee, 2017), which have been applied in RL to highlight input relevance (He et al., 2021; Carbone, 2020; Beechey et al., 2023).

---

[1]https://rish-av.github.io/bexrl

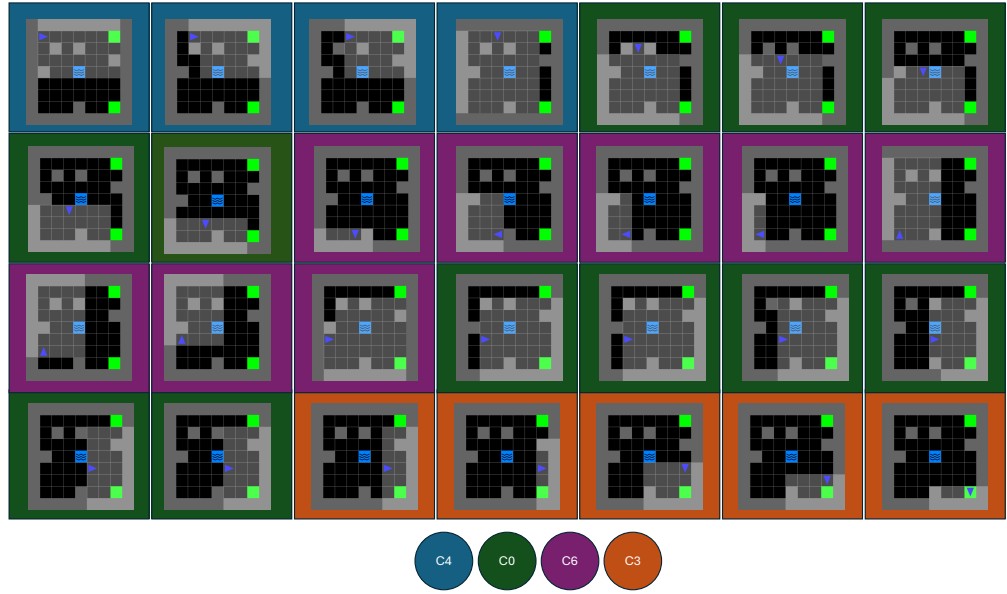

Figure 1: Representative trajectory from the MiniGridTwoGoalsLava environment, annotated with behavior cluster assignments shown via colored backdrops (c0, c1, etc, best viewed at 200%). The full trajectory is shown left-to-right, top-to-bottom, with segments assigned to distinct behavior clusters. For example, c2 captures exploration, c0 corresponds to lava traversal, and c3 involves goal approach. This segmentation reveals temporally extended patterns in agent behavior. Descriptions for all clusters are summarized in Table 1, and visual samples from each cluster along are provided in Appendix C.

Although effective for local analysis, these methods operate at the level of single timesteps and provide only snapshot-level insight. As a result, they fail to capture long-term structure in agent behavior.

Other approaches expand the scope by examining how past events influence agent decisions. Causal graph models (Madumal et al., 2020) aim to explain actions by modeling dependencies between decisions and their outcomes. Counterfactual techniques such as EDGE (Guo et al., 2021) identify causal features by intervening on state representations. StateMask (Cheng et al., 2023) takes a different approach, learning to mask parts of the input sequence that are less relevant for predicting returns. While these methods can highlight important moments or compress decision histories, they do not capture higher-level behavioral patterns or recurring strategy components. Their focus remains on local explanations rather than structured descriptions of what the agent is doing over time.

Trajectory-based methods aim to capture longer-term context when explaining agent decisions. For example, Deshmukh et al. (2024) attribute decisions to full trajectories retrieved from offline data that may have influenced the agent. While this provides more temporal information than single-step explanations, it treats each trajectory as a whole, without distinguishing between different phases of behavior. As a result, actions taken during exploration, goal-seeking, or recovery may all be attributed to the same trajectory, making it hard to tell which specific behavior was responsible for a given decision.

In this work, we propose a behavior-centric framework for post-hoc attribution in RL. Rather than assigning decisions to individual features, single states, or entire trajectories, we segment rollouts into temporally coherent clusters that capture recurring patterns in agent behavior, as shown in the annotated MiniGridTwoGoalsLava rollout in Figure 1. Agent decisions are then attributed to these discovered behavior segments, which are identified in an unsupervised manner and validated through visualization and human evaluation. This behavior-level attribution enables analysis of how different strategies influence outcomes. For example, in an autonomous driving scenario, it can reveal that unsafe lane changes consistently occur during a specific merging behavior. Such insights support systematic identification of policy failures and can inform safety auditing, strategy refinement, and monitoring of policy drift.

Our main contribution is a behavior-centric explanation framework that attributes agent decisions to unsupervised behavior segments, rather than isolated states or full trajectories. The method operates entirely offline and does not require access to rewards or the underlying policy model. By clustering sequences of state-action pairs, we produce behavior-level interpretations of agent decisions that reflect recurring execution patterns. To our knowledge, this is the first approach to explain RL policies through unsupervised discovery of behavior segments. We evaluate the framework on four offline RL benchmarks and show that it identifies coherent behavioral clusters, enables faithful and interpretable attributions, and outperforms trajectory-level baselines in both quantitative metrics and human studies.

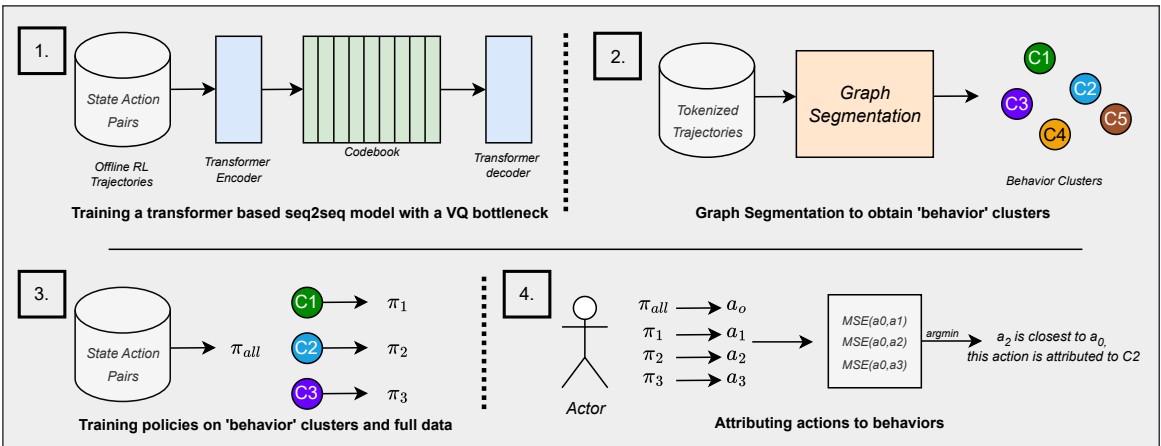

Figure 2: Overview of the proposed behavior attribution framework. A transformer-based Vector Quantized Variational Autoencoder (VQ-VAE) is trained on state-action sequences to produce discrete latent codes, which are used to tokenize trajectories. These tokenized trajectories are then segmented via graph clustering to discover distinct behavior clusters. Policies are trained on each cluster as well as the full dataset. During attribution, actions taken by the original policy are compared to those from each cluster policy, and attribution is made based on minimal mean squared error (MSE), assigning actions to the most behaviorally aligned cluster. A causal mask ensures that encoding respects the temporal structure of trajectories. The transformer architecture is described in detail in Appendix B.1.

## 2 Related Work

**Behavior Discovery in RL.** Structured behavior discovery has been studied extensively in RL to support temporal abstraction, planning, and policy reuse. The Options framework (Sutton et al., 1999; Precup, 2000) formalized temporally extended actions. Later works (Bacon et al., 2017; Harb et al., 2017; Jin et al., 2022) proposed learning such abstractions from data using intrinsic or extrinsic signals. Unsupervised skill discovery approaches (Eysenbach et al., 2018; Achiam et al., 2018; Villecroze et al., 2022) maximize diversity or rely on nonparametric clustering to identify latent behaviors. In the offline setting, OPAL (Ajay et al., 2021) and Diffuser (Janner et al., 2022) utilize learned behavioral embeddings to support planning or imitation.

LAPO (Schmidt & Jiang, 2024) and Genie (Bruce et al., 2024) extend these ideas by discretizing behavior via VQ-based latent spaces for downstream control or simulation. However, such methods are designed for learning compact action spaces or optimizing policies, not for attributing observed behavior. In contrast, we apply sequence-level discretization to state-action trajectories for the purpose of discovering coherent behavior segments. These are subsequently segmented via a graph-based approach to yield interpretable behavior units that support post-hoc explanation in a reward-free, offline setting.

**Explainability in RL.** A key objective in Explainability in Reinforcement Learning (XRL) is to understand why an agent takes a particular action. Feature-level methods attribute decisions to salient input

regions using gradient-based techniques (Greydanus et al., 2018; Zahavy et al., 2017) or perturbation-based importance (Puri et al., 2019; Iyer et al., 2018). Others adopt techniques like LIME (Ribeiro et al., 2016) and SHAP (Lundberg & Lee, 2017) to explain decisions in reinforcement learning by attributing importance to input features (Beechey et al., 2023; Zhang et al., 2022).

Other works focus on attributing importance to individual transitions. EDGE (Guo et al., 2021) uses counterfactuals to identify which state features caused a particular action. StateMask (Cheng et al., 2023) learns to mask non-critical observations without degrading performance, enabling step-level saliency. RICE (Cheng et al., 2024) extends this idea by identifying critical states and reinitializing the agent from them to improve training. AIRS (Yu et al., 2023) and Liu et al. (Liu et al., 2023) attribute decisions to specific moments using reward gradients or visual cues. These methods effectively highlight pivotal decisions, but typically focus on isolated steps or episodes and do not uncover structured behavioral patterns that persist across trajectories. Moreover, many of them rely on reward signals or environment access, whereas our approach is fully offline and reward-free, identifying behavioral structure from trajectories alone.

Several approaches instead operate at the level of trajectory segments or full rollouts. HIGHLIGHTS (Amir & Amir, 2018) selects representative snippets to summarize policy behavior but does not explain individual actions. AOC (Sun et al., 2023) retrieves similar decisions from a corpus to justify actions, but formulates explanation as policy construction rather than post-hoc analysis. Deshmukh et al. (Deshmukh et al., 2024) are most closely related to our work, as they attribute actions to full trajectories retrieved from offline data. However, their method treats each trajectory as a single unit, which can bundle together multiple distinct behaviors and reduce attribution granularity. In contrast, we decompose trajectories into temporally coherent, recurring behavior modules and associate each action with its corresponding behavior. This allows for structured, behavior-level attribution grounded in patterns that recur across episodes.

Some approaches aim to explain agent behavior by modeling its internal structure or learning simplified surrogates. Causal frameworks (Madumal et al., 2020; Pawlowski et al., 2020) build structural models to answer contrastive or counterfactual queries, revealing why specific actions were taken. These methods can offer deep insight into decision rationale, but often rely on access to predefined or learned causal graphs, which may be hard to obtain in complex domains. Distillation-based approaches (Coppens et al., 2019; Verma et al., 2019) approximate policies using decision trees or symbolic programs. While interpretable, they focus on replicating outputs rather than exposing the underlying decision process, offering limited insight into the factors or abstractions that drive decision-making.

**Latent Discretization in RL** Discrete representations have proven useful for structuring policies and simplifying planning in RL. VQ-VAEs (Oord et al., 2017) have been adopted for compressed modeling (Hafner et al., 2020), action quantization (Luo et al., 2023), and hierarchical decision-making (Nachum et al., 2018). LAPO (Schmidt & Jiang, 2024) recovers symbolic latent actions via inverse dynamics, and Genie (Bruce et al., 2024) learns behavior tokens for autoregressive simulation. Unlike these methods, we do not use discretization to build a control policy. Instead, we leverage it to recover a graph-structured segmentation over trajectories, enabling post-hoc analysis and modular attribution of learned behaviors.

**Evaluation in XRL** Surveys on explainable reinforcement learning (Cheng et al., 2025; Vouros, 2022) emphasize the lack of unified evaluation standards. While fidelity, interpretability, and human alignment are commonly used, most work focuses on feature saliency or trajectory summaries. Few methods address behavior-level explainability. Our approach introduces structured, temporally grounded explanations via unsupervised behavior discovery. We evaluate it through fidelity scores, cluster coherence, and human preference, offering a distinct contribution to the XRL landscape.

## 3 Method

We propose a three-stage framework for post-hoc behavior discovery and attribution in offline reinforcement learning. First, we train a transformer-based VQ-VAE to discretize state-action trajectories into latent codes that capture temporally extended behavioral motifs. Next, we construct a behavior graph using these codes and apply spectral clustering to identify coherent behavior segments. Finally, we attribute actions from a

pretrained policy to the discovered behaviors by comparing them with cluster-specific behavior models. This framework enables structured, interpretable analysis of agent behavior directly from offline data.

### 3.1 Behavior Discovery

Behavior discovery is at the core of our framework, where we identify and segment meaningful sub-trajectories from offline RL data. The process begins with a transformer-based Vector Quantized Variational Autoencoder (VQ-VAE) trained in a sequence-to-sequence fashion, producing discrete latent codes for each time step, which are later used for segmentation.

The encoder processes sequences of state-action pairs $\{(s_t, a_t), (s_{t+1}, a_{t+1}), \dots, (s_{t+k}, a_{t+k})\}$, where $s_t \in \mathbb{R}^n$ (or $s \in \mathbb{R}^{n \times n}$ for image-based observations), and $a_t \in \mathbb{R}^m$. Positional encodings and causal masking ensure temporally valid and autoregressive representation learning. The encoder outputs latent vectors $z_t \in \mathbb{R}^d$, which are then discretized via a learned codebook $\mathcal{C} = \{c_1, c_2, \dots, c_N\}$, by computing:

$$q(z_t) = \arg\min_{c_i \in \mathcal{C}} \|z_t - c_i\|^2 \tag{1}$$

The VQ-VAE loss combines codebook and commitment objectives:

$$\mathcal{L}_{\text{vq}} = \mathbb{E}\left[\|z_t - \text{sg}(c_q)\|^2\right] + \mathbb{E}\left[\|\text{sg}(z_t) - c_q\|^2\right] \tag{2}$$

where $\text{sg}(\cdot)$ denotes stop-gradient. The decoder reconstructs future states $\{\hat{s}_{t+1}, \hat{s}_{t+2}, \dots\}$ using only past information. The total training objective is:

$$\mathcal{L} = \mathcal{L}_{\text{recon}} + \alpha \mathcal{L}_{\text{vq}}, \quad \text{where} \quad \mathcal{L}_{\text{recon}} = \mathbb{E}\left[\sum_{t=1}^{k} \|s_t - \hat{s}_t\|^2\right] \tag{3}$$

Using state-action pairs instead of only states is key to discovering behavior-centric latent representations. Prior works like LAPO (Schmidt & Jiang, 2024) and Genie (Bruce et al., 2024) have shown that when only states are used as input, the learned latents tend to capture action-like primitives, effectively forming a latent action space. In contrast, by including actions alongside states, the model learns representations that reflect what the agent actually did in each context, leading to latents that correspond to temporally extended behaviors rather than individual action decisions. Similar insights are seen in Deshmukh et al. (2024), which also incorporate action and reward information to better isolate the behaviors underlying agent decisions.

Although the VQ-VAE provides per-timestep discretization, the autoregressive attention in transformers leads to delay in token shifts relative to behavior change. Further, minor input perturbations can yield different latent codes for the same behavior. To address this, we apply a graph-based segmentation that smooths the raw discretization into coherent behavior segments.

### 3.2 Behavior Segmentation

To uncover structured behaviors from offline RL data, we segment trajectory sequences using a graph-based approach over latent tokens. Each token is a discrete representation of a state-action chunk, obtained via vector quantization in the behavior discovery stage. The segmentation graph captures both how frequently these tokens transition into one another and how similar they are in latent space, capturing both temporal and structural coherence.

Formally, we define a graph $G = (V, E)$, where each node $v_i \in V$ corresponds to a codebook token $c_i$, and edge weights encode a hybrid of transition dynamics and latent similarity:

$$w_{ij} = (1 - \lambda) \cdot \text{Count}(c_i \rightarrow c_j) + \lambda \cdot \|c_i - c_j\|_2^2 \tag{4}$$

Here, $\text{Count}(c_i \rightarrow c_j)$ is the normalized frequency of transitions in the dataset, while the Euclidean term penalizes large jumps in latent space. The hyperparameter $\lambda \in [0, 1]$ controls the trade-off between temporal

continuity and spatial clustering. We empirically ablate $\lambda$ in Appendix A and show that moderate values consistently yield better segmentation. This hybrid weighting is necessary: transition-only graphs may over-segment due to noise or repeated transitions between distinct behaviors, while distance-only graphs may merge temporally distinct behaviors with similar representations. Combining both allows the method to recover temporally extended, recurring behavioral motifs.

We apply spectral clustering to segment the resulting graph. First, we compute the symmetric similarity matrix $S$ as $S_{ij} = w_{ij} + w_{ji}$, and construct the unnormalized Laplacian:

$$L = D - S, \quad \text{where} \quad D_{ii} = \sum_j S_{ij} \tag{5}$$

The top $k$ eigenvectors of $L$ embed the graph nodes into a lower-dimensional space, where connected components become geometrically separable. The number of clusters $k$ is determined via eigengap analysis. Each cluster corresponds to a distinct behavior, and each timestep in the dataset is labeled accordingly.

We also compare this segmentation approach with alternatives, such as clustering directly in latent space (e.g., $k$-means) or over raw state-action pairs. As discussed in Appendix A, spectral clustering consistently outperforms these alternatives by yielding more temporally stable and semantically meaningful clusters.

| Cluster # | HalfCheetah-medium-v2 | MiniGridTwoGoalsLava | Seaquest-mixed-v0 | Pen-expert-v1 |
|---|---|---|---|---|
| 0 | Jump prep using hind legs | Crossing lava | Exploration variety | Initial Grasp |
| 1 | Running on hind legs | Running into obstacle/wall | Idle Gameplay | Final Pose Adjustment |
| 2 | Leaping forward | Exploration variety | Transition phase | Manipulating pen |
| 3 | Gait transition | Approaching goals | Moving downwards | Stable hand |
| 4 | Bounding with accel. | Forward exploration | Agent killed | Noisy cluster |
| 5 | Bounding | Stuck at wall | Random firing | Pose adjustment |
| 6 | Fall recovery | Continuous motion | Moving right | – |
| 7 | Accel. leap | Transition phase | Moving left | – |
| 8 | Running on front leg | Grid exploration | Exploration variety | – |
| 9 | Running on hind legs | Transition phase | Filling oxygen | – |
| 10 | – | Corner exploration | – | – |
| 11 | – | Clean exploration | – | – |

Table 1: Interpretable behavior clusters discovered across four benchmark environments. Descriptions are based on visual inspection of representative samples randomly chosen per cluster. Visual samples and detailed descriptions for each cluster across all environments are available in Appendix C.

### 3.3 Attributing Actions to Behaviors

We attribute policy actions to discovered behavior clusters to explain agent decisions in terms of learned behavioral motifs. A policy $\pi$ is trained on the full dataset. For each behavior cluster $k$, we train a behavior cloning model $M_k$ using its trajectory segments.

In continuous action spaces, attribution uses MSE:

$$k^* = \arg\min_k \text{MSE}(a, \hat{a}_k), \quad \text{MSE}(a, \hat{a}_k) = \frac{1}{d} \sum_{i=1}^d (a_i - \hat{a}_{k,i})^2 \tag{6}$$

In discrete domains, attribution uses cross-entropy:

$$k^* = \arg\min_k \mathcal{L}_{\text{CE}}(p_a, p_k), \quad \mathcal{L}_{\text{CE}}(p_a, p_k) = -\sum_{j=1}^C p_{a,j} \log(p_{k,j}) \tag{7}$$

This process yields behavior-level explanations for each action, helping clarify which learned behavior motif best accounts for a given decision.

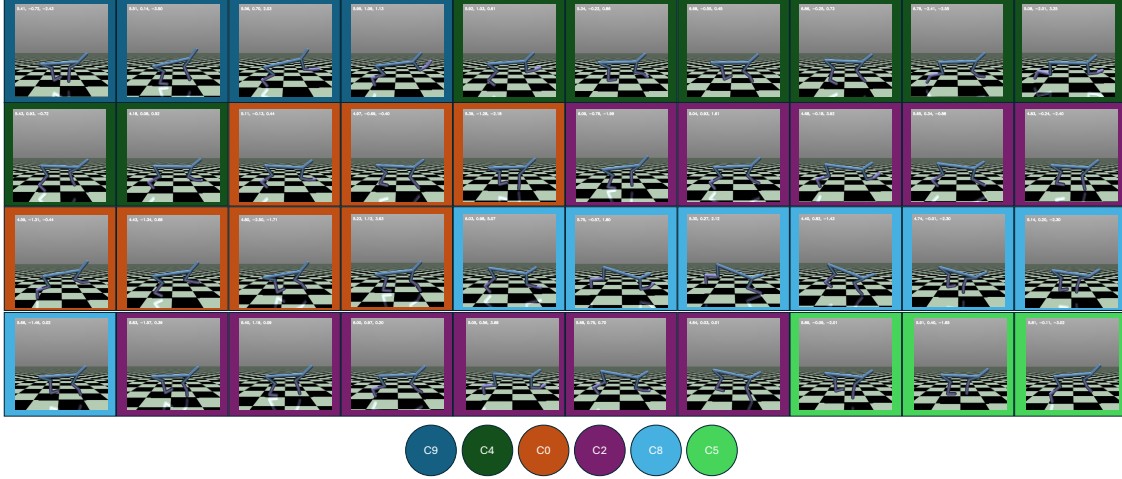

Figure 3: A sample sequence from HalfCheetah with cluster labels visualized through colored backdrops for each frame. For instance, Cluster 0 involves hind-leg loading before a leap, Cluster 8 captures running on front leg, and Cluster 2 corresponds to high speed forward leaps. Full cluster definitions are in Table 1; Sample sequences from each cluster can be found in Appendix C.

## 4 Experiments

We evaluate the effectiveness of our framework for behavior discovery and attribution using three benchmark environments from D4RL(Fu et al., 2020)—halfcheetah-medium-v2, pen-expert-v1—as and seaquest-mixed-v0 from D4RL-atari repository (Takuseno, 2025) as well as a custom environment, MiniGridTwoGoalsLava, based on the MiniGrid suite. These environments span a broad spectrum of RL challenges: halfcheetah-medium-v2 involves high-dimensional continuous locomotion; seaquest-mixed-v0 is a discrete-action Atari domain with dense visual input and long-horizon objectives; pen-expert-v1 requires precise dexterous manipulation with high-dimensional proprioceptive and object-centric observations; and MiniGridTwoGoalsLava supports goal-conditioned navigation with clearly interpretable behavior switches. The datasets for the D4RL environments were collected using partially trained SAC (Haarnoja et al., 2018)(halfcheetah), DQN (Mnih et al., 2015) (seaquest), and scripted expert policies (pen), as described in Fu et al. (2020), and reflect varying levels of sub-optimality to simulate realistic offline conditions. The dataset for MiniGridTwoGoalsLava was generated by us using a PPO policy trained to approximately 30–40% task success. Our pipeline uses a transformer-based VQ-VAE model to discretize state-action sequences into latent codes, which are used to construct a behavior graph over time. Spectral clustering is then applied to obtain segmented trajectories, as described in Section 3.2. To assess the robustness of our method, we conduct a broad ablation study varying the codebook size, number of clusters, graph construction parameters (e.g., $\lambda$ for balancing transition vs. latent similarity), and architecture components. Full results are presented in Appendix A, with hyperparameters listed in Appendix B.1.

### 4.1 Behavior Discovery and Segmentation

We evaluate whether the behavior clusters discovered by our framework reflect temporally coherent and semantically meaningful segments across four offline RL environments. The segmentation is based on latent codes obtained from a VQ-VAE trained on state-action sequences, post-processed using a spectral clustering algorithm over a graph constructed from transition frequency and latent similarity. This hybrid formulation enables the discovery of segments that are both structurally consistent and behaviorally distinct.

As summarized in Table 1, our method isolates a range of interpretable behaviors across environments. In MiniGridTwoGoalsLava, clusters include exploring near walls, crossing lava, and approaching goals. In seaquest-mixed-v0, we observe clusters corresponding to oxygen refilling, directional movement, and repet-

itive firing. For halfcheetah-medium-v2, the clusters capture different locomotion modes such as bounding, leaping, and gaits involving instability or braking.

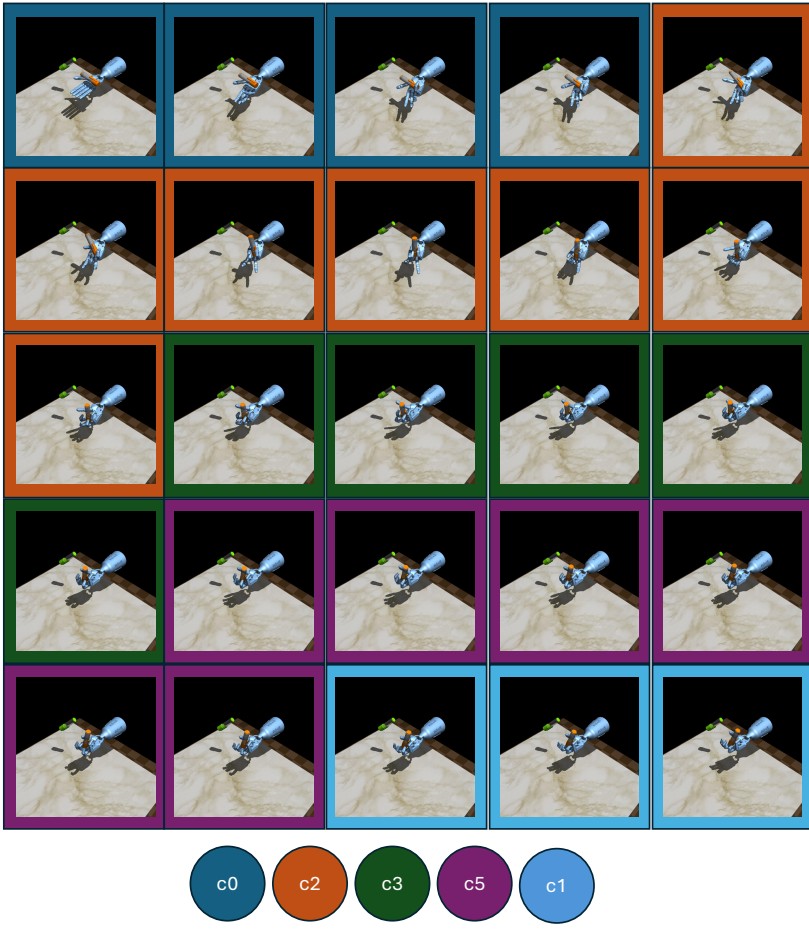

Figure 4: Visualization of a sample trajectory from pen-expert-v1, displayed with a frameskip of 2. Cluster assignments are indicated by the colored backdrops for each frame, with corresponding cluster IDs (c0, c1, c2, c3, c5) shown below the trajectory. Full details of cluster definitions can be found in Table 1 and representative samples from each cluster can be found in Appendix C. This image is best viewed at 200% zoom.

Figures 1, 5, 3 & 4 provide rollout-level visualizations of the discovered clusters, highlighting how behavior boundaries align with high-level shifts in strategy or movement. Cluster assignments typically remain stable within consistent behaviors and change sharply at clear behavioral transitions, often at action-level granularity. Across episodes, we observe that similar behavior types are consistently grouped together—suggesting that the framework recovers reusable, high-level behavior patterns rather than overfitting to surface-level differences in input.

While these visualizations offer qualitative validation, we also conducted a human study (described next) to quantitatively assess the interpretability of the discovered behavior clusters. Participants described sample segments from different clusters and provided confidence scores, allowing us to measure agreement and semantic coherence.

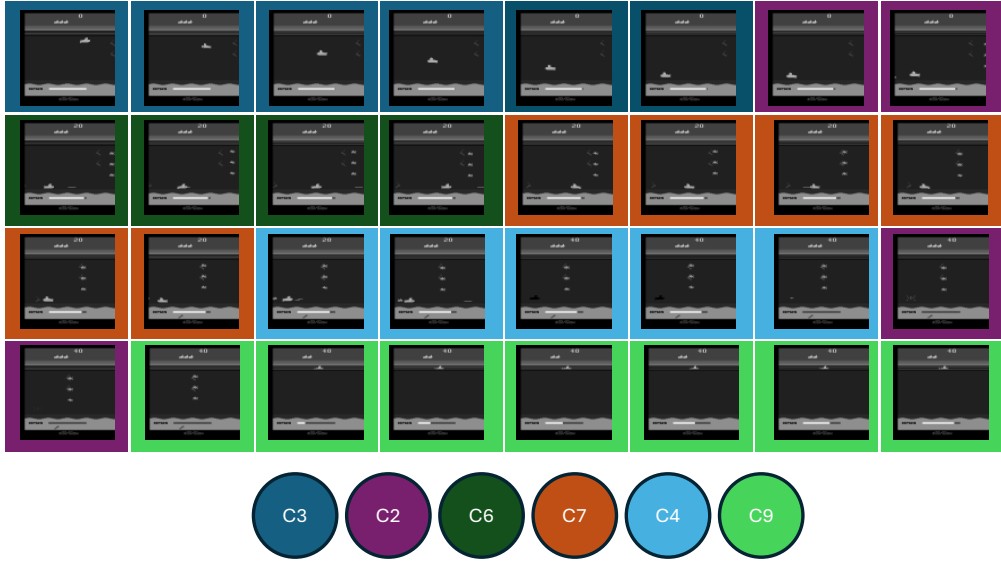

Figure 5: Sample from Seaquest-mixed-v0 with cluster labels visualized through colored backdrops for each frame. Cluster 3: moving down, Cluster 6: moving right, Cluster 9: oxygen refill. See Table 1 for more. We further present sample sequences from each cluster in Appendix C.

### 4.1.1   Human Study: Behavior Interpretability

To evaluate the semantic coherence of our discovered clusters, we conducted a two-part human study. Part A was designed to assess whether clusters correspond to behaviors that are interpretable and describable by humans.

For each of the four environments, we randomly sampled three behavior clusters. For each cluster, three representative trajectory segments were selected and shown to 15 participants who had prior familiarity with the tasks. Participants were asked to (1) describe the cluster behavior in natural language, and (2) rate their confidence on a 1–5 scale. We computed the average pairwise cosine similarity between the textual descriptions using MiniLM sentence embeddings, after subtracting a shared embedding representing the general environment context. Higher intra-cluster similarity indicates consistent human interpretations of the behavior.

| Environment | Cluster1 | Cluster2 | Cluster3 | AvgCosine | AvgConfidence |
|---|---|---|---|---|---|
| seaquest-mixed-v0 | 0.68 | 0.69 | 0.63 | 0.67 | 5.00 |
| MiniGridTwoGoalsLava | 0.68 | 0.64 | 0.58 | 0.63 | 4.33 |
| halfcheetah-medium-v2 | 0.61 | 0.65 | 0.61 | 0.62 | 5.00 |
| pen-expert-v1 | 0.64 | 0.61 | 0.57 | 0.61 | 4.33 |

Table 2: Intra-cluster cosine similarities and average confidence scores for human-written descriptions of sampled behavior clusters across four environments. Higher cosine values indicate greater semantic agreement among annotators, reflecting stronger behavioral coherence within clusters. For comparison, the average cosine similarity between randomly selected descriptions from different clusters was **0.42**.

The consistently high cosine similarities and confidence scores across environments suggest that our clusters are semantically meaningful and interpretable for most participants. Participants often used similar phrases (e.g., "approaching goal", "crossing obstacle", "leaping forward") to describe different samples from the same cluster, supporting the behavioral consistency captured by our framework.

## 4.2 Behavior Attribution

To validate the relevance of discovered behaviors, we perform action-level attribution by training a separate behavior cloning (BC) model for each cluster. Details about the BC models and their training can be found in Appendix B.2. These models serve as local approximators for how each behavior typically responds in given states. For any observed action, we identify the most likely behavior by comparing the output of these models to the agent's actual action.

In continuous control environments (e.g., *halfcheetah-medium-v2*), we compute the mean squared error (MSE) between the action predicted by each cluster model and the ground truth action. The cluster minimizing this error is selected as the behavior attribution. In discrete domains (e.g., seaquest-mixed-v0, MiniGridTwoGoalsLava), we compare action probability distributions using cross-entropy.

This attribution mechanism consistently assigns decisions to plausible behavior clusters across environments. For example, in seaquest-mixed-v0, sequences involving upward swimming and rapid shooting are reliably attributed to distinct clusters. In halfcheetah-medium-v2, the attribution clearly distinguishes between behaviors like high-speed bounding and slower gait transitions. These results suggest that the segmented behaviors not only reflect semantic structure but also support faithful explanation of policy behavior. Attribution examples are visualized in Figure 6.

### 4.2.1 Human Study: Attribution Quality

To evaluate whether our behavior-level attributions align with human intuition, we conducted Part B of the human study. In this setting, participants ($n = 15$) were shown a short context window consisting of preceding and following frames surrounding a specific agent action. They were then asked to select which of four video segments best explained that decision.

The four options were: (1) the behavior segment attributed by our method, (2) a trajectory segment retrieved using the method of Deshmukh et al. (2024), and (3–4) randomly sampled segments of similar length from other behavior clusters. To avoid trivial selections, none of the presented options contained the query segment itself. The goal was to assess which independent segment best captured the behavioral pattern underlying the given action.

Importantly, all video options were presented without labels, metadata, or trajectory statistics to minimize cognitive bias arising from cluster identity or video length. This setup encourages participants to rely solely on the semantics of the action context when judging which segment provides the most plausible and behaviorally aligned explanation.

| Environment | %Ours | %Deshmukh et al. (2024) | %Random |
|---|---|---|---|
| pen-expert-v1 | 85.7 | 14.3 | 0.0 |
| MiniGridTwoGoalsLava | 61.9 | 28.5 | 9.5 |
| seaquest-mixed-v0 | 68.4 | 31.6 | 0.0 |
| halfcheetah-medium-v2 | 81.3 | 13.9 | 4.7 |

Table 3: Human preference scores across environments for the segment attributed by our method, the trajectory-level baseline from Deshmukh et al. (2024), and random segments. Our method was most frequently selected as the best explanation in all environments. The largest margins were observed in pen-expert-v1 and halfcheetah-medium-v2, highlighting the benefit of fine-grained, behavior-centric attribution.

These results suggest that human evaluators consistently find our behavior-level attributions to be more plausible and semantically aligned with the observed action than those derived from full-trajectory retrieval methods. This supports our intuition that behavior segments serve as more targeted and reusable explanatory units, especially in domains where agent behavior varies continuously or includes overlapping strategies. Additional details regarding this study are provided in Appendix D.

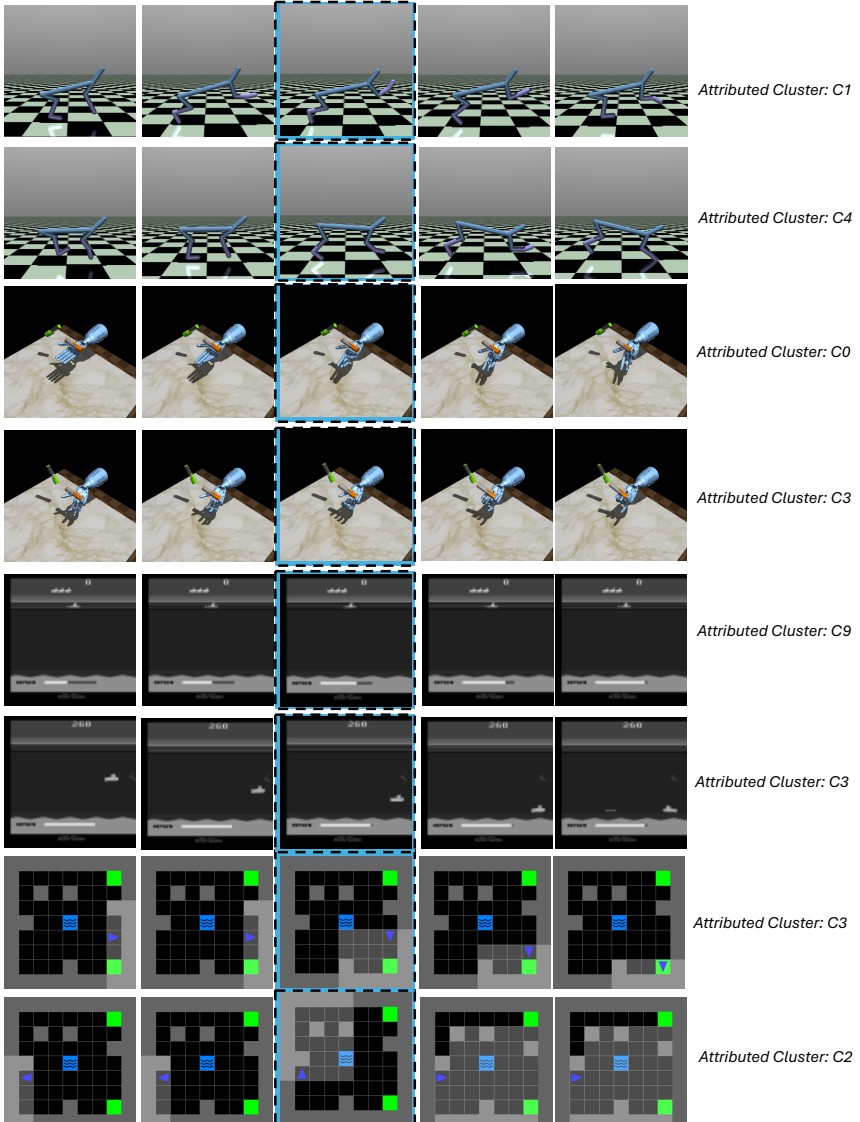

Figure 6: Attribution results for various environments. The attribution, as described in Section 3.3, is performed for the middle frame in each sequence (indicated with a dashed border). To provide better temporal context, both preceding and succeeding frames are shown. The assigned behavior cluster is noted on the right. Across all environments, the assigned clusters align well with the observed behavior. For example, in HalfCheetah, cluster 1 corresponds to hind-leg running, while cluster 4 captures bounding accelerating motion. Similarly in Sequest, cluster 9 corresponds to filling oxygen and cluster 3 corresponds to downward motion. Similar consistency is also seen for MiniGrid and Pen.

## 4.3 Quantitative Evaluation: Cluster Soundness and Structural Quality

To complement the qualitative and human studies, we quantitatively evaluate our discovered behavior clusters using two perspectives: (1) how consistently each cluster captures policy-aligned behavior (*cluster soundness*), and (2) how structurally coherent and well-separated the clusters are in latent space (*structural quality*). These evaluations aim to assess whether the clusters are not only interpretable but also statistically meaningful.

**Cluster Soundness via Fidelity Score.** To assess whether our discovered clusters represent distinct and consistent behavioral modes, we compute the Average Fidelity Score (AFS): the mean prediction error

between a behavior cloning (BC) model trained per cluster and the original policy. A low AFS implies that the cluster encapsulates a coherent policy fragment. To assess the distinctiveness of each cluster, we compare this to a random baseline where actions are assigned to clusters uniformly at random (Rand-Ours), maintaining the same cluster sizes.

Table 4 shows that the AFS gap between our clusters and the random assignment is substantial, indicating that each of our clusters captures a unique behavioral pattern. In contrast, for Deshmukh et al. (2024), the fidelity scores of their clusters and their random baseline (Rand-Deshmukh et al. (2024)) are often very close. This suggests their segments are not strongly aligned with distinct decision-making behavior, and that their clusters may be less behaviorally meaningful.

| Environment | Ours | Rand-Ours | Deshmukh et al. (2024) | Rand-Deshmukh et al. (2024) |
|---|---|---|---|---|
| halfcheetah-medium-v2 | 0.36 | 2.29 | 0.05 | 0.09 |
| seaquest-mixed-v0 | 0.59 | 0.98 | 0.13 | 0.17 |
| MiniGridTwoGoalsLava | 0.47 | 0.68 | 0.09 | 0.10 |
| pen-expert-v1 | 0.16 | 0.49 | 0.13 | 0.20 |

Table 4: Average Fidelity Score (AFS) for our method, random cluster assignments, and the trajectory-level baseline from Deshmukh et al. (2024). AFS is computed as the error between the main policy (trained on the full dataset) and the output of the behavior model corresponding to a given cluster. Scores are averaged over 200 randomly selected actions from 20 episodes. In our method, the larger gap between attributed and random cluster scores (Ours vs. Rand-Ours) indicates that the discovered clusters capture clearly distinct behaviors. In contrast, the small gap observed for the baseline suggests its clusters are less behaviorally differentiated.

**Structural Cluster Quality.** In addition to fidelity, we evaluate the structural properties of the discovered clusters in the latent space using two standard unsupervised clustering metrics. The *Silhouette Score* (SS) captures how well each latent token fits within its assigned cluster versus the next-best alternative—higher values indicate better separation. The *Davies-Bouldin Score* (DB) compares intra-cluster dispersion to inter-cluster separation—lower values are better. Together, these metrics reflect whether the latent structure of the clustering is compact and distinct.

As shown in Table 5, our method generally produces clusters with better separation (higher SS) and lower within-cluster variance (lower DB) than Deshmukh et al. (2024), particularly in continuous and high-dimensional settings such as halfcheetah-medium-v2 and seaquest-mixed-v0. In MiniGridTwoGoalsLava, the baseline scores slightly better due to its LSTM-based encoder specialized for discrete grid settings, whereas our model uses a uniform transformer backbone across all environments. Despite this, our approach yields more interpretable segments (as seen in AFS and human studies), suggesting that structural metrics alone may not fully capture behavior-level coherence.

| Environment | Silhouette Score ↑ | | Davies-Bouldin ↓ | |
|---|---|---|---|---|
| | Ours | Deshmukh et al. (2024) | Ours | Deshmukh et al. (2024) |
| halfcheetah-medium-v2 | 0.24 | 0.14 | 1.68 | 2.13 |
| seaquest-mixed-v0 | 0.21 | 0.12 | 1.88 | 2.19 |
| MiniGridTwoGoalsLava | 0.37 | 0.48 | 1.44 | 1.19 |
| pen-expert-v1 | 0.20 | 0.20 | 1.47 | 1.32 |

Table 5: Structural cluster quality using Silhouette and Davies-Bouldin scores across four environments. Our method yields more compact and separated clusters in continuous control (halfcheetah-medium-v2) and high-dimensional image-based tasks (seaquest-mixed-v0). In grid-based domains such as MiniGridTwoGoalsLava, the LSTM-based encoder of the baseline offers better numerical clustering scores, though these do not always translate to semantically meaningful behaviors (see Table 4 and human studies).

## 5 Limitations

The proposed framework, while effective in discovering and attributing behaviors, has certain limitations. In highly stochastic environments such as halfcheetah-medium-v2, noisy or frequent behavior transitions can result in overlapping segments that are less clearly delineated. While most clusters correspond to interpretable behaviors, a few exhibit ambiguity, making it difficult to precisely characterize their semantic content. Moreover, our graph-based segmentation relies on hand-crafted similarity measures and traditional spectral clustering, which may not optimally adapt to diverse environments. Future work could investigate graph neural network-based segmentation methods that learn task-specific similarity structures and provide more robust behavior partitioning.

## 6 Conclusion

We introduced a post-hoc framework for discovering and attributing behavior-level patterns in offline reinforcement learning. By combining sequence-level discretization through a transformer-based VQ-VAE with graph-based segmentation via spectral clustering, our method extracts interpretable, temporally extended behaviors from raw trajectories. Attribution is performed by matching policy actions to behavior-specific models, enabling fine-grained explanations of decisions. Evaluations across diverse domains, including locomotion, manipulation, and Atari-style control, demonstrate that the discovered segments align with semantically meaningful behaviors and offer improved interpretability over trajectory-level baselines. While current results are limited to offline settings, future extensions could explore applications in debugging, auditing, and safe deployment of RL agents in real-time systems.

## 7 Acknowledgment

We sincerely thank NSERC, CIFAR, Digital Research Alliance of Canada & Denvr Datacloud for providing resources to complete this project.

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

# A Appendix A: Ablation Studies

## A.1 Alternative Clustering Strategies

We evaluate the effectiveness of our clustering approach by comparing it against simpler alternatives. Specifically, we assess whether learned representations are necessary for high-quality segmentation and whether spectral clustering offers benefits over standard methods like K-means. These comparisons help isolate the contribution of each component—representation learning and clustering algorithm—to the overall quality of behavior segmentation.

### A.1.1 Clustering Raw State-Action Pairs

To evaluate the necessity of learning representations, we compare our method against a baseline where clustering is performed directly on raw state-action pairs, without any VQ-VAE encoder. As shown in Table 6, this results in significantly worse clustering performance across all environments. The clusters tend to be noisy and inconsistent, with poor separation between distinct behaviors and no temporal coherence. These results demonstrate the importance of using a representation model like VQ-VAE to transform trajectories into discrete, behavior-centric codes that capture higher-level structure.

Table 6: Clustering performance when applied directly to raw state-action pairs. Absence of VQ-based representation results in weak and noisy clusters.

| Environment | Silhouette Score ↑ | Davies-Bouldin Score ↓ |
|---|---|---|
| halfcheetah-medium-v2 | 0.20 | 1.97 |
| pen-expert-v1 | 0.13 | 2.23 |
| MiniGridTwoGoalsLava | 0.10 | 2.14 |
| seaquest-mixed-v0 | 0.03 | 8.88 |

### A.1.2 Spectral Clustering vs K-Means

We also compare spectral clustering, our default segmentation method, with standard K-means applied to the learned VQ-VAE embeddings. As shown in Table 7, spectral clustering outperforms K-means in both Silhouette Score and Davies-Bouldin Score across environments. Unlike K-means, spectral clustering leverages the global structure of the transition graph and is better suited for identifying temporally smooth and behaviorally meaningful clusters.

Table 7: Comparison between spectral clustering and K-means applied to latent embeddings. Spectral clustering produces more temporally coherent and behaviorally meaningful clusters.

| Environment | Method | Silhouette Score ↑ | Davies-Bouldin Score ↓ |
|---|---|---|---|
| halfcheetah-medium-v2 | Spectral | 0.24 | 1.75 |
| halfcheetah-medium-v2 | K-Means | 0.20 | 1.97 |
| MiniGridTwoGoalsLava | Spectral | 0.37 | 1.44 |
| MiniGridTwoGoalsLava | K-Means | 0.30 | 2.05 |
| pen-expert-v1 | Spectral | 0.20 | 1.47 |
| pen-expert-v1 | K-Means | 0.17 | 1.82 |
| seaquest-mixed-v0 | Spectral | 0.22 | 2.55 |
| seaquest-mixed-v0 | K-Means | 0.21 | 2.65 |

## A.2 Lambda Factor

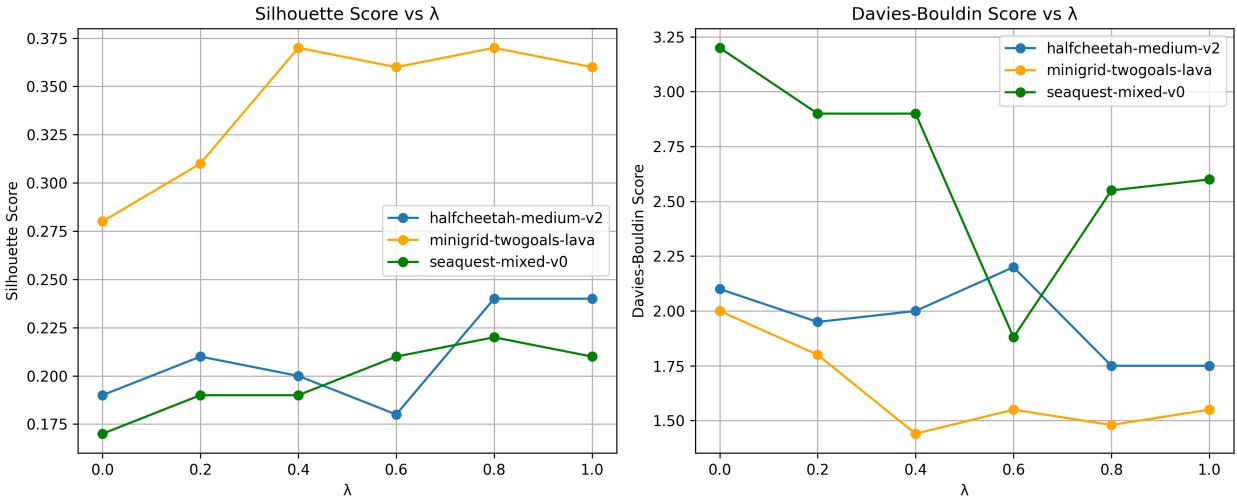

Figure 7: Clustering quality metrics across $\lambda$ values. $\lambda = 0$ uses only transition frequency; $\lambda = 1$ uses only latent-space proximity. Best performance typically occurs at intermediate values.

We study the effect of the $\lambda$ parameter that controls the trade-off between transition frequency and latent-space proximity in edge weighting. Figure 7 shows clustering quality metrics (Silhouette and Davies-Bouldin scores) across $\lambda$ values. A balanced weighting ($\lambda \approx 0.3$ to $0.5$) typically yields the best results, confirming that both temporal and semantic structure are important for stable segmentation.

## A.3 Number of Codebooks

We evaluate the sensitivity of our method to the number of codebook entries in the VQ-VAE, as this hyperparameter governs both representational capacity and interpretability. Using too few codebook entries limits the expressiveness of the latent space, resulting in poor reconstruction and insufficient coverage of behavior diversity. On the other hand, very large codebooks can lead to sparsely used or entirely inactive (dead) codes, reducing interpretability and increasing latent fragmentation.

To select an appropriate codebook size, we consider two metrics: (1) normalized reconstruction loss, which assesses how well the latent space encodes input sequences, and (2) codebook occupancy, defined as the proportion of codebook entries used at least once over the final 10% of training steps. This smoothing avoids instability from early-phase underutilization.

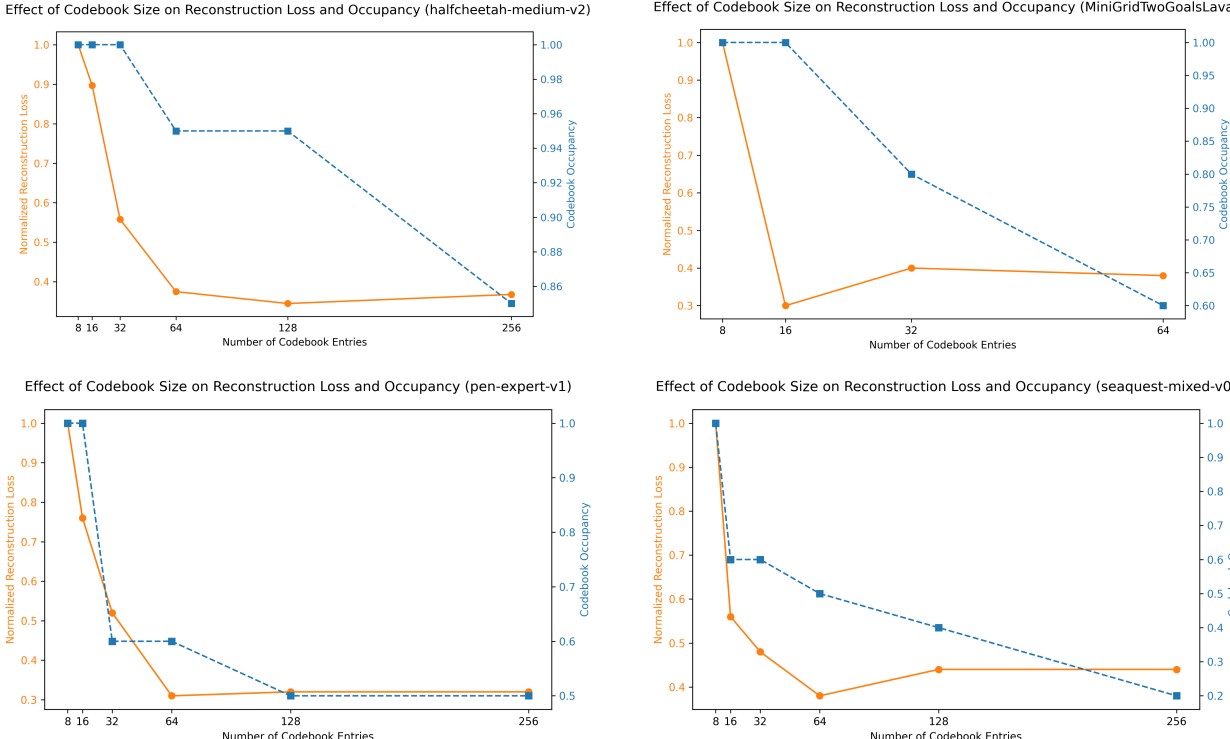

Figure 8: Effect of codebook size on normalized reconstruction loss (orange, lower is better) and codebook occupancy (blue, higher is better) across four environments. Increasing the number of codebooks improves expressiveness but may reduce code utilization. Our method achieves stable performance when balancing these two metrics. In most domains, codebook sizes in the 64–128 range yield good trade-offs. Notably, in MiniGridTwoGoalsLava, a smaller codebook of size 16 offers both low reconstruction loss and high occupancy, suggesting that optimal codebook size can be environment-dependent.

# B  Hyperparameters & Training

## B.1  Transformer-based VQ-VAE Architecture

## B.2  Behavior Cloning Models

To support attribution and post-hoc analysis, we trained separate Behavior Cloning (BC) models on the clustered state-action data. Each cluster had a sufficient number of state-action samples to allow reliable model fitting. However, due to noise in the clustering process, some individual state-action pairs were occasionally isolated from their surrounding behavior segments. To improve consistency, we applied a smoothing step that reassigned such outliers to the most frequent cluster label among adjacent steps in the trajectory.

For training, we used a 3-layer multilayer perceptron (MLP) for environments with state-based observations, and a 3-layer convolutional neural network (CNN) for environments with image-based inputs. The learning rate and other training details match those in Table 8.

# C  Cluster Visualization

To better interpret the discovered behavior clusters, we randomly sample 20 trajectory segments assigned to each cluster and analyze them based on visual inspection. For each cluster, we then describe the most common recurring patterns observed across the sampled segments. This process helps illustrate the typical

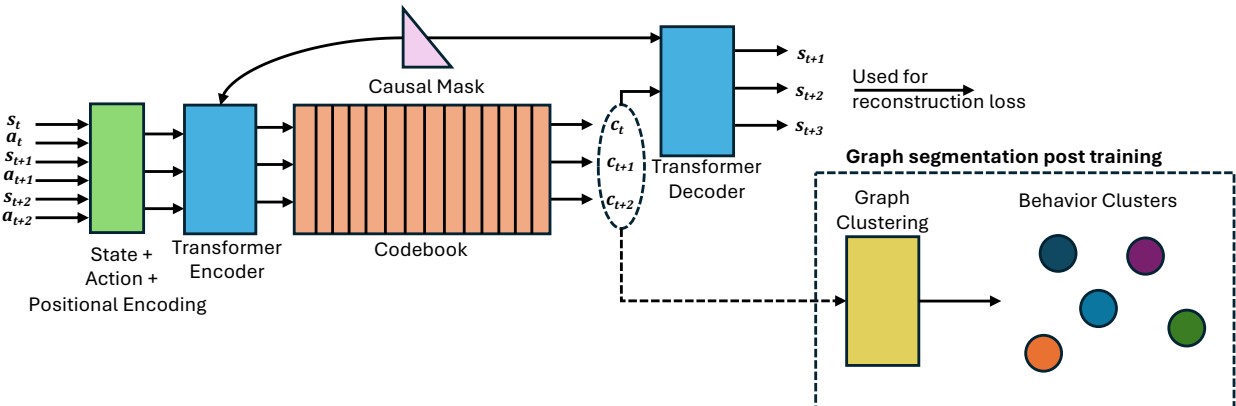

Figure 9: Overview of the behavior discovery and segmentation pipeline. A transformer-based VQ-VAE is trained to encode sequences of state-action pairs into discrete latent codes using a codebook. The encoder applies causal masking to ensure autoregressive prediction, and the decoder reconstructs future states to optimize a reconstruction loss. After training, the resulting latent codes are used to construct a transition graph, which is segmented using spectral clustering to produce semantically coherent behavior clusters. Refer to Table 8 for full details on the architecture and training.

| Hyperparameter | halfcheetah-medium-v2 | MiniGrid-TwoGoalsLava | seaquest-mixed-v0 | pen-expert-v1 |
|---|---|---|---|---|
| Learning Rate (LR) | $1 \times 10^{-4}$ | $1 \times 10^{-4}$ | $1 \times 10^{-4}$ | $1 \times 10^{-4}$ |
| Sequence Length (seq_len) | 50 | Variable (max 40) | 30 | 30 |
| Batch Size | 64 | 32 | 64 | 32 |
| Number of Codes | 128 | 16 | 64 | 64 |
| Embedding Dimension | 128 | 128 | 128 | 128 |
| Combination Param ($\lambda$) | 0.75 | 0.45 | 0.6 | 0.6 |
| Num Epochs | 50 | 50 | 50 | 50 |
| Optimizer | Adam | Adam | Adam | Adam |
| LR Scheduler | Linear decay | Linear decay | Linear decay | Linear decay |
| Teacher Forcing | Linear decay to 0 | Linear decay to 0 | Linear decay to 0 | Linear decay to 0 |
| Transformer Heads | 4 | 4 | 4 | 4 |
| Encoder/Decoder Layers | 4 | 2 | 4 | 4 |
| Transformer Hidden Dim | 128 | 128 | 128 | 128 |
| Frame Skip | – | – | 4 | – |
| Hardware | A100 GPU | A100 GPU | A100 GPU | A100 GPU |

Table 8: Hyperparameter settings for all four environments. Note that sequence length is truncated or padded for environments with short episodes, and GPU used for all experiments is NVIDIA A100.

behaviors captured by each cluster and highlights the consistency of behavioral motifs discovered by our approach.

## D   Human Study Details

To assess the interpretability and plausibility of the discovered behavior clusters, we conducted a two-part human study covering both cluster semantics (Part A) and behavior attribution (Part B). The design for our attribution study (Part B) was inspired by the human evaluation protocol in Deshmukh et al. (2024), where participants are asked to choose the trajectory that best explains an agent's action. This appendix provides

Table 9: Number of state-action pairs per cluster used for BC training across environments.

| Environment | C0 | C1 | C2 | C3 | C4 | C5 | C6 | C7 | C8 | C9 | C10 | C11 |
|---|---|---|---|---|---|---|---|---|---|---|---|---|
| halfcheetah-medium-v2 | 11393 | 10510 | 10815 | 10780 | 5183 | 9646 | 6153 | 11951 | 2891 | 3122 | – | – |
| seaquest-mixed-v0 | 9710 | 3321 | 2894 | 7471 | 6692 | 9980 | 7081 | 7763 | 3510 | 7398 | – | – |
| MiniGridTwoGoalsLava | 942 | 743 | 1217 | 932 | 936 | 746 | 938 | 432 | 1215 | 481 | 1114 | 1316 |
| pen-expert-v1 | 8264 | 4213 | 9711 | 4232 | 2677 | 8819 | – | – | – | – | – | – |

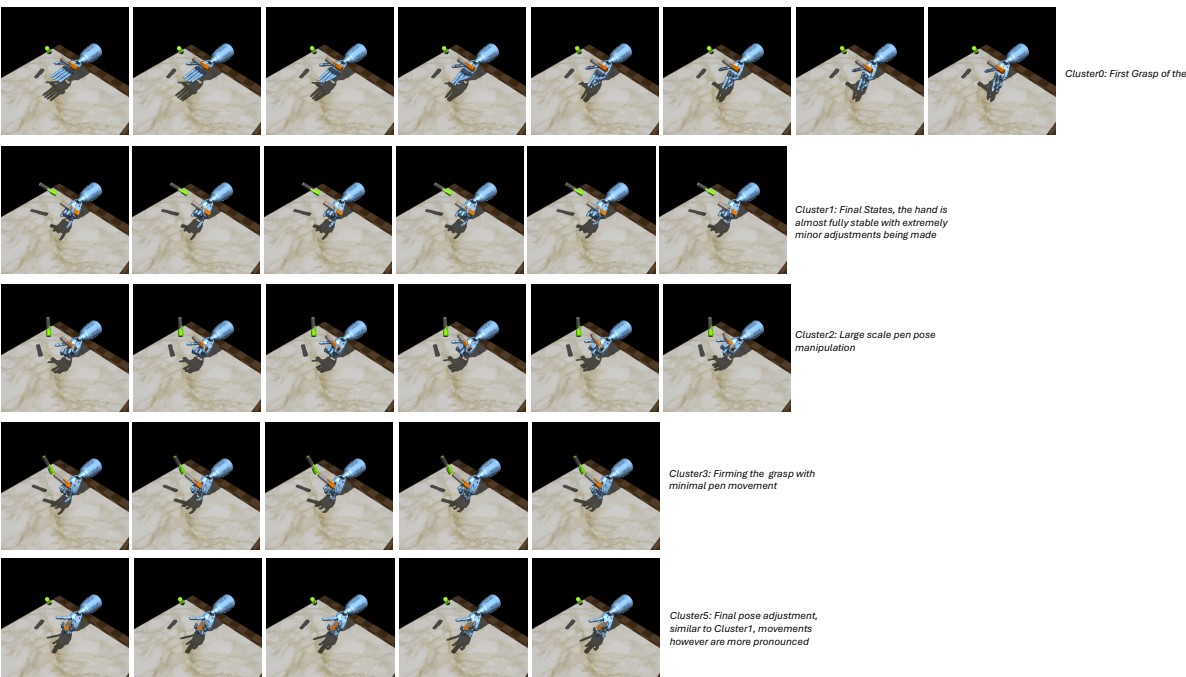

Figure 10: Set of all pen-expert-v1 clusters with their detailed descriptions. Behaviors exhibit different stages of pen manipulation. Cluster 4 is skipped since there are no representative samples for it as its quite noisy.

additional details about the design, instructions, and interface used in each part. Figures 15 and 14 show example questions presented to participants for the interpretability and attribution studies, respectively. All participants had prior familiarity with the environments and were not informed of the clustering method or any ground truth labels.

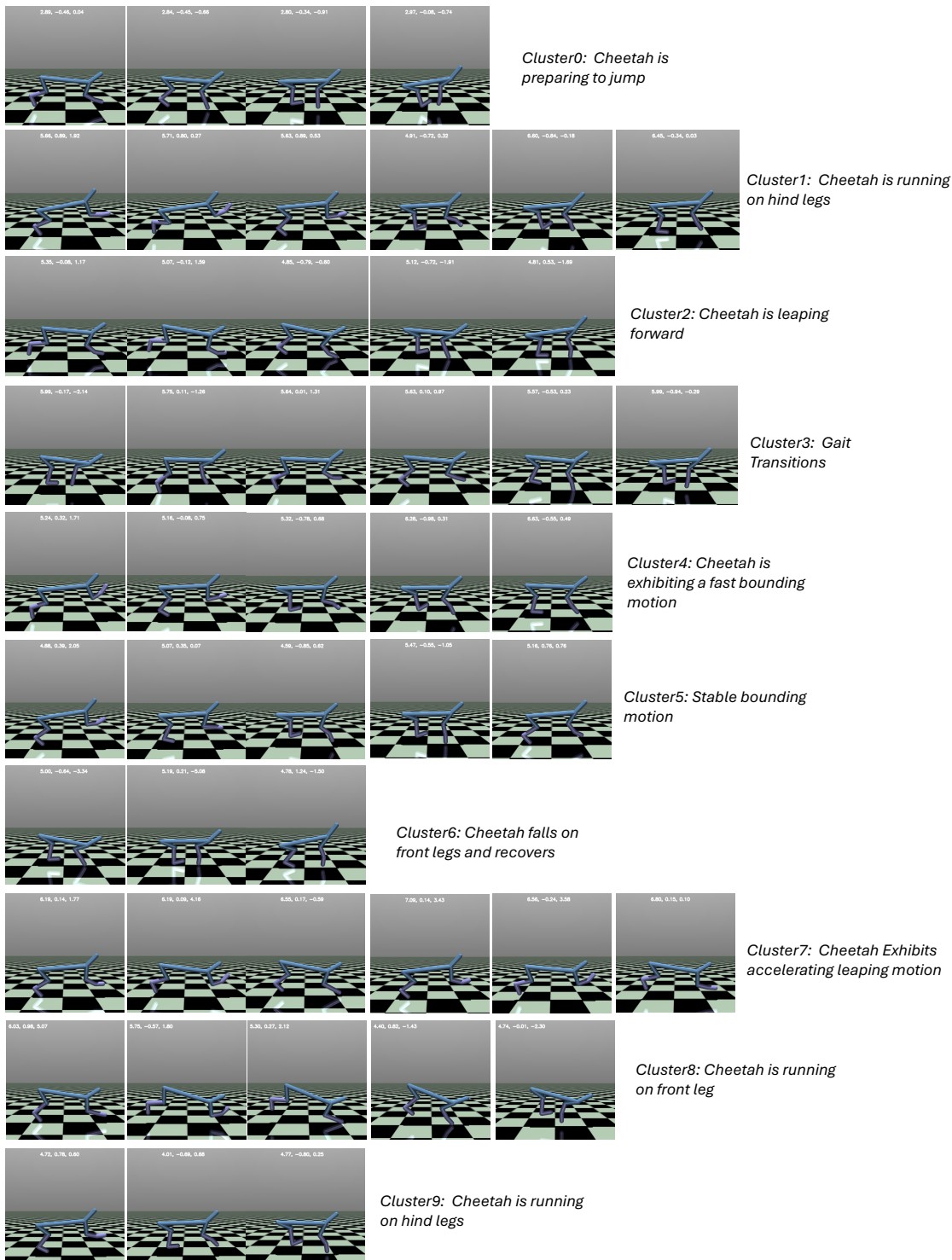

Figure 11: Set of all halfcheetah-medium-v2 clusters and their detailed descriptions. Most discovered behaviors show some form of motion. Some clusters are easy to inspect and understand visually but some are difficult given the nature of the environment. The numbers in the images velocities of the cheetah which provides a broader context.

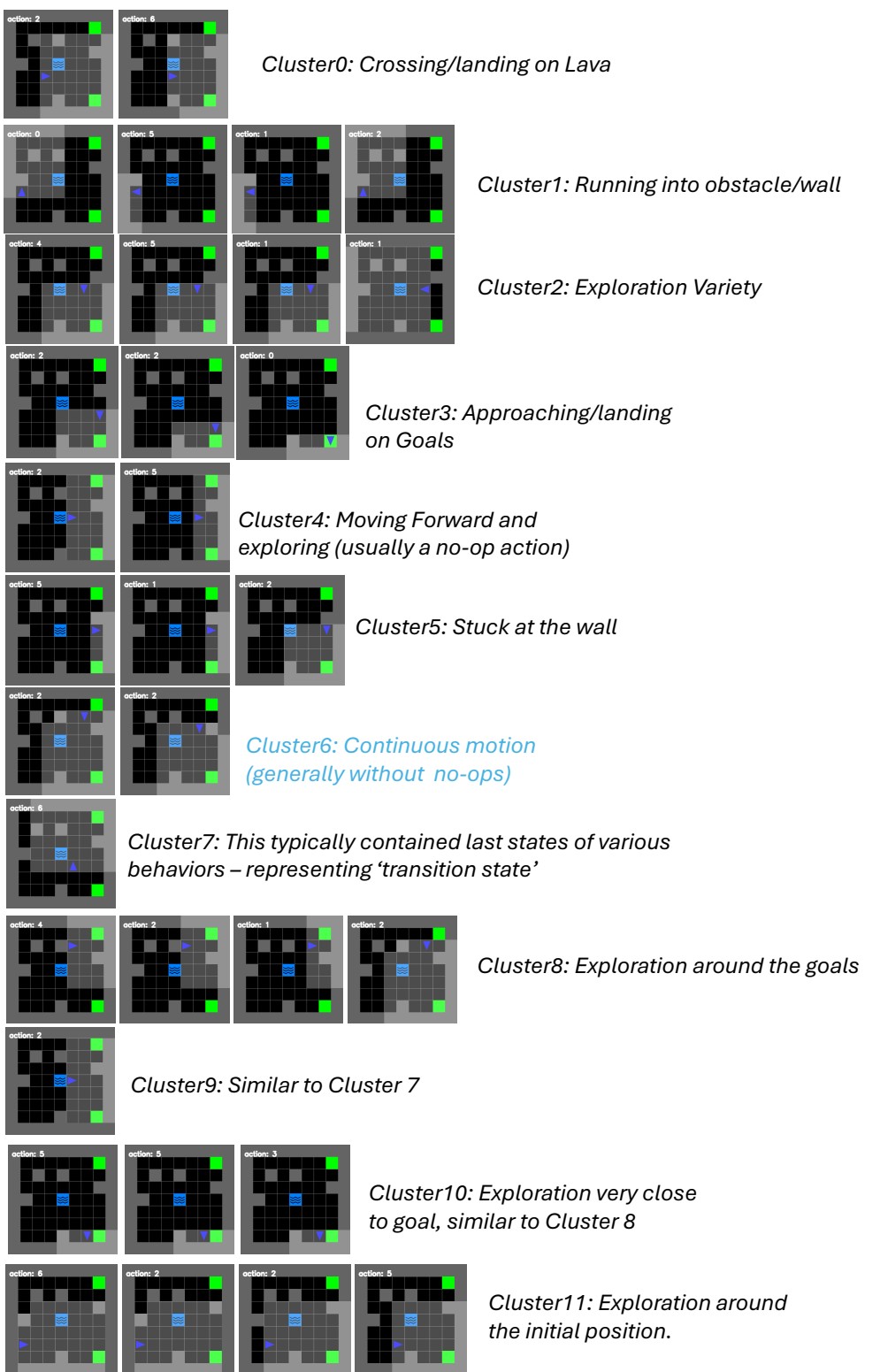

Figure 12: Set of all MiniGridTwoGoalsLava clusters with their corresponding description. Behaviors have good diversity, exploratory behaviors of distinct varieties are also discovered using this mechanism.

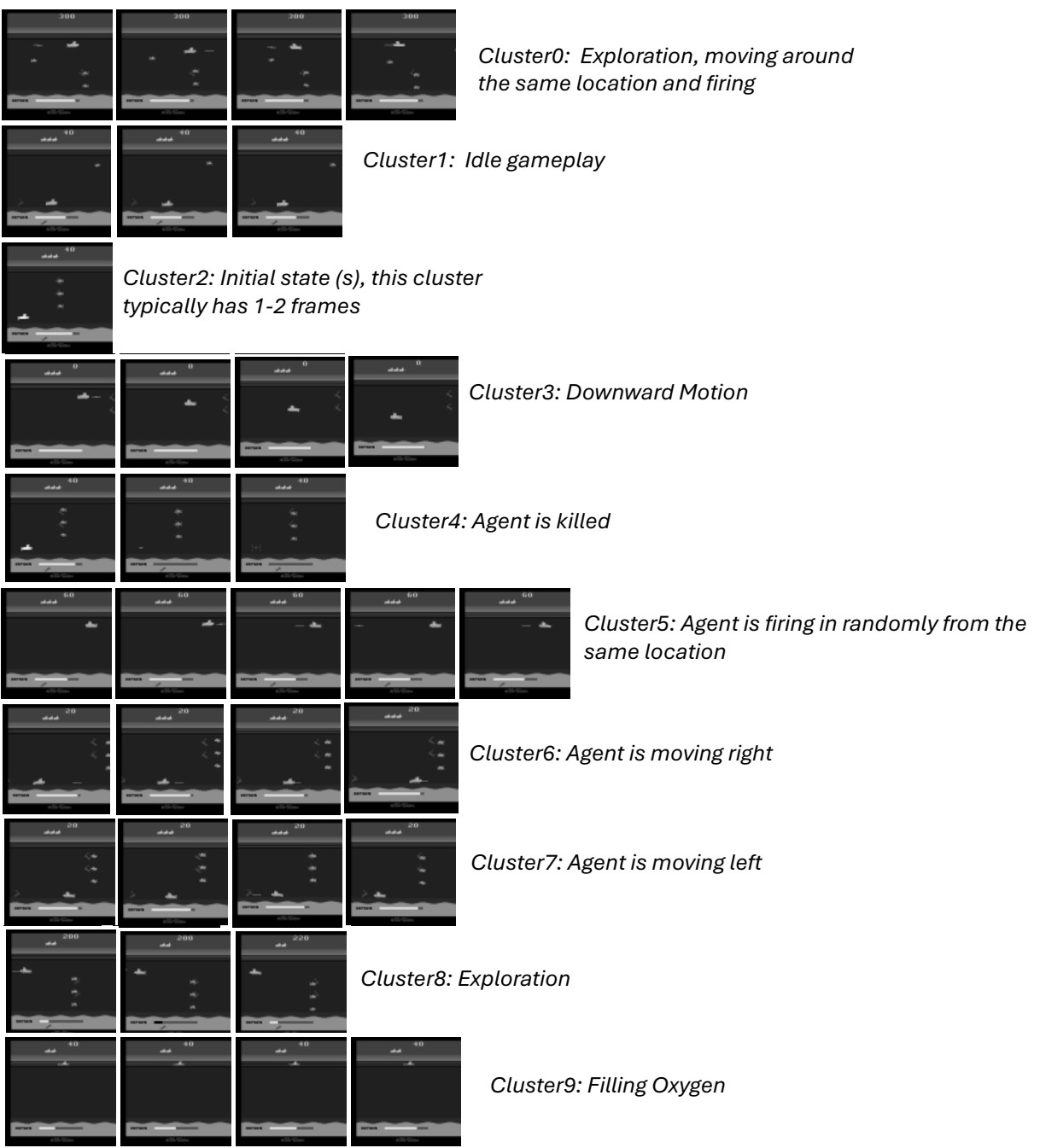

Figure 13: Set of all seaquest-mixed-v0 clusters and their detailed descriptions. There are some meaningful behaviors as well as some exploratory clusters where it is hard to exactly define the behavior. This is expected since the data is from a suboptimal agent, thus exploratory behavior is expected.

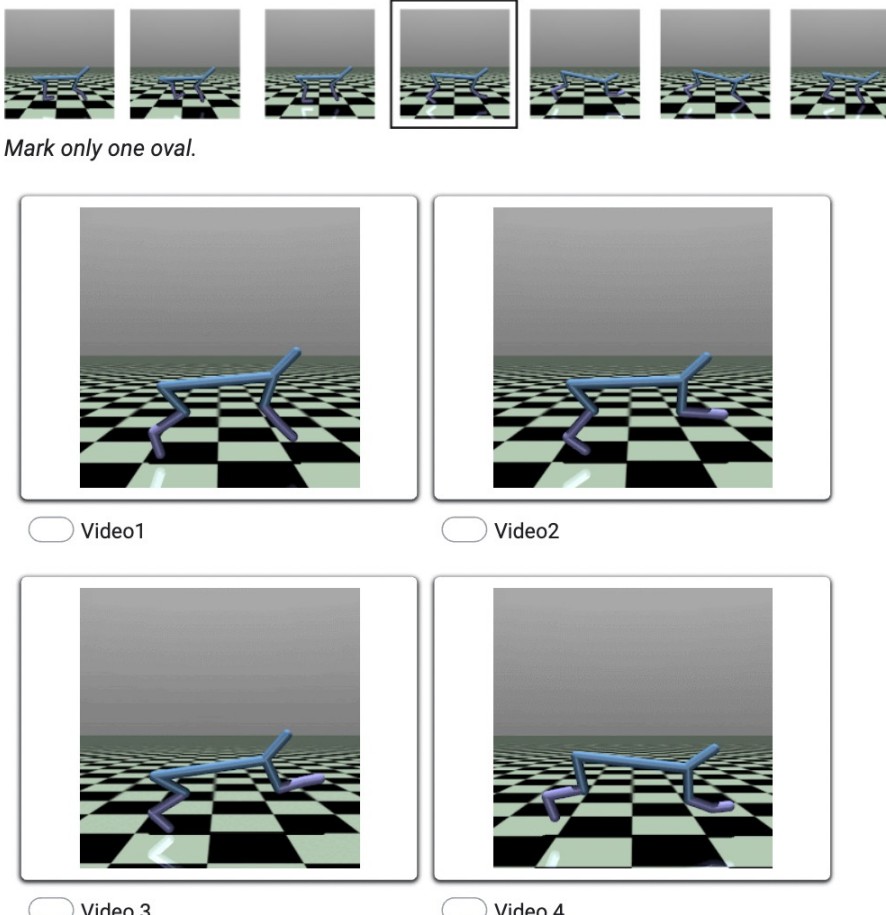

Figure 14: Sample question from Human Study Part B (Attribution Quality). Participants are shown a short context window centered on a specific agent action (top row) and are asked to rank four candidate video segments (bottom) based on how well they explain the observed action. The query context is excluded from the options, encouraging participants to generalize behavioral patterns. The four options include the segment attributed by our method, a trajectory retrieved via Deshmukh et al. (2024), and two random segments.

2.  Describe this in 1-2 lines. There are multiple rows, you can explain each independently *
    or jointly.

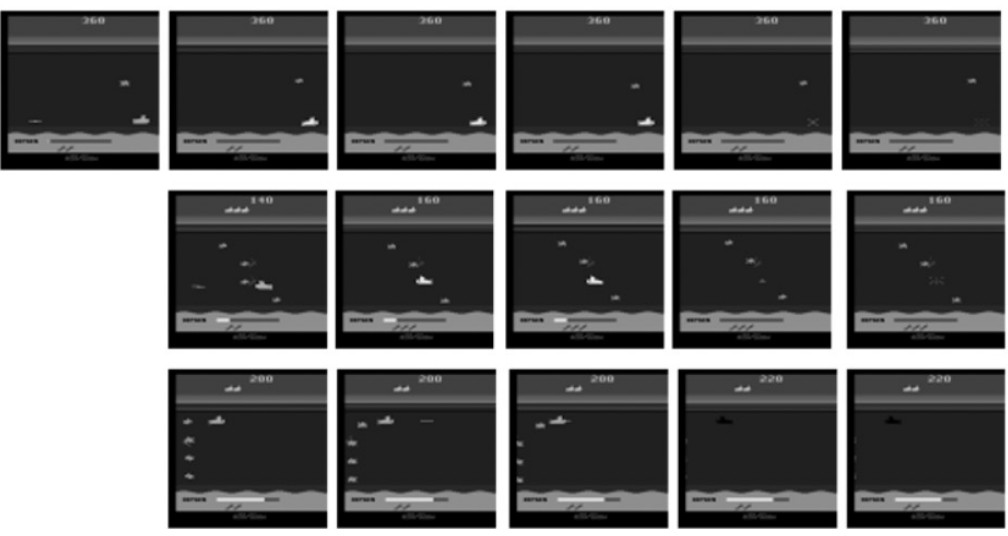

3.  What is your confidence about the description? *

    *Mark only one oval.*

         1   2   3   4   5

    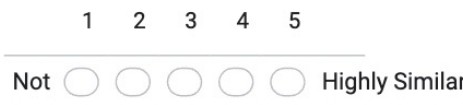

    Not   ◯ ◯ ◯ ◯ ◯   Highly Confident

4.  How similar do you think are each of these sequences? *

    *Mark only one oval.*

         1   2   3   4   5

    Not   ◯ ◯ ◯ ◯ ◯   Highly Similar

Figure 15: Sample question from Human Study Part A (Behavior Interpretability). Participants are shown a set of trajectory segments sampled from the same behavior cluster and asked to describe the behavior in 1–2 lines. They are then asked to rate their confidence in the description and judge how similar the sequences appear. These ratings are used to assess semantic coherence and human alignment of the discovered clusters.

