# OpenReview forum: "Behaviour Discovery and Attribution for Explainable Reinforcement Learning"
_TMLR — Accepted by TMLR_

### Review · Reviewer_Ttp7 · 2025-07-03

**Summary Of Contributions:**

This paper proposes a three-stage pipeline for explaining the behaviors of offline RL policies: (1) fit a transformer-based VQ-VAE to trajectories in the offline dataset, encoding trajectories $s_0, a_0, \dots, s_t, a_t$ into discrete codes $c_i$ by reconstructing future trajectories ${s_{t+1}, a_{t+1}, \dots, s_T, a_T}$ (2) build a transition graph using discrete codes as vertices and edge weights determined by the transition probability and similarity, then apply spectral clustering to obtain $K$ clusters, (3) train $K$ behavior-mode polices using data from each cluster. This pipeline can be used to (1) segment trajectories into clusters, and (2) behavior attribution by comparing the predicted action of a policy to the output of behavior-mode policies. Experiments show more preferred human ratings and higher clustering scores compared to prior trajectory-level behavior attribution methods.

**Audience:**

Yes

**Broader Impact Concerns:**

The paper does not have ethical implications that warrant a Broader Impact Statement.

**Claims And Evidence:**

No

**Requested Changes:**

1. Provide clearer visualizations in figures 1, 3, 4, 5, 6 (e.g. color-code the clusters and add colored backdrops to the frames) or use more conspicuous examples.
2. Explain the human studies in more detail. The second human study is not very clear. For example, what does it mean for a human to select a video to explain a behavior? Wouldn't the exact same video be the best explanation?
3. Why limit the setting to RL? This method doesn't make assumptions about the policy learning method. You can similarly apply this to behavior cloning policies and datasets.

**Strengths And Weaknesses:**

**Strengths**
1. The proposed method is novel and a reasonable approach for explaining the behaviors of RL policies.
2. Qualitative visualization and human ratings validates the effectiveness of the proposed method.

**Weaknesses**
1. The figures in the paper are hard to interpret. Figures 1, 3, 4, 5, and 6 are supposed to show trajectories segmented into behavior clusters, but just by looking at the figures without any context, I can't tell the clusters apart. The captions help, but they induce confirmation bias.
2. The main experiments involve qualitative visualizations and human studies, which are not quite convincing.
3. The problem setting of behavior attribution has limited usefulness for the community.

---

> ### Author Response · Authors · 2025-07-12
> **Response to reviewer Ttp7**
>
> We thank the reviewer for their constructive feedback and for helping make our work better. We answer the concerned questions below:
>
> ### Clairty on Experimental Results
>
> We would like to emphasize the validity of our experiments and highlight the quantitative and comparative evidence that supports our claims. Our evaluation is intentionally multifaceted, combining quantitative metrics, qualitative visualizations, and human studies to provide a holistic and robust validation of our framework.
>
> * **Quantitative Validation**: Our claims are not based solely on visualizations or human intuition. We provide strong **quantitative evidence** that our discovered behavior clusters are more coherent and meaningful than the trajectory-level clusters from our baseline.
>     * The **Average Fidelity Score (AFS)** in Table 4 shows a substantial gap between our clusters and a random assignment, indicating that our clusters represent distinct and consistent policy fragments. In contrast, the baseline's clusters are much closer to their random counterpart, suggesting they are less behaviorally differentiated.
>     * The **structural clustering metrics** (Silhouette and Davies-Bouldin Scores) in Table 5 further show that our method produces more compact and well-separated clusters, especially in high-dimensional continuous control and image-based tasks.
>
> * **Extensive Human Studies**: We believe our human-centered evaluation is a primary strength of the paper. Our study design is more extensive than that of our baseline, Deshmukh et al. (2024), which was conducted on a single grid-world environment. We evaluate across **all four diverse environments**, providing stronger evidence of generalization. Furthermore, our human preference study is a standard and effective methodology for evaluating the utility of XAI systems [1,2,3]. In our study (Part B), participants were asked to choose which explanation best captured the behavioral pattern of a query action, a common paradigm in user-centric evaluation. This comparative setup directly assesses the quality of our behavior-level explanations against trajectory-level and random ones, with our method being strongly preferred.
>
> ### **Clearer Visualizations**
>
> Thank you for this suggestion. We have revised the figures in the paper to improve clarity. As requested, we have enhanced the visualizations in Figures 1, 3, 4, 5, and 6 with clearer, color-coded backdrops to make the behavior segments more distinct.
>
> Regarding the use of more "conspicuous" examples, we intentionally do not hand-pick specific trajectories to avoid selection bias. Our process involves randomly sampling trajectories to provide an honest representation of our method's performance. That said, we did randomly sample several new trajectories for our revision. While the results for `HalfCheetah` and `Pen` were consistent with what was already shown, we found clearer representative examples for `MiniGrid` and `Seaquest` and have updated those figures accordingly.
>
> ### **Human Study Details**
>
> We have expanded the description of our human studies in the paper to provide more clarity. To directly address the reviewer's request, we have added a new **Appendix D** that includes the exact questions and screenshots presented to the human evaluators for both parts of the study.
>
> Our use of a comparative preference study is a standard methodology in XAI evaluation. As surveyed by Rong et al. [1], numerous studies directly compare different explanation methods to assess their effectiveness [2, 3]. Our study follows this established practice.
>
>
> For the specific question, "Wouldn't the exact same video be the best explanation?", this is an excellent point. To ensure the study was not a simple matching task, we **explicitly excluded the query video itself from the four choices presented to the user**. This design forces participants to generalize and select the video that best captures the *underlying behavioral pattern* of the query action, which is a much stronger test of our attribution quality. We have clarified this important detail in Section 4.2.1 of the paper.
>
>
> ### **References**
> [1] Rong, Y., et al. (2023). Towards Human-centered Explainable AI: A Survey of User Studies for Model Explanations.
> [2] Buçinca, Z., et al. (2020). Proxy tasks and subjective measures can be misleading in evaluating explainable AI systems. *IUI*.
> [3] Sixt, L., et al. (2022). Do users benefit from interpretable vision? a user study, baseline, and dataset. *ICLR*.
>
>
> ### **Note: Continued in the next comment for Broader Applicability Beyond RL**

---

> > ### Author Response · Authors · 2025-07-12
> > **Continued from Previous Comments**
> >
> > ### **Broader Applicability Beyond RL**
> >
> > The reviewer raises an excellent point. Our method is, in essence, not strictly limited to reinforcement learning. It could indeed be applied to other behavioral datasets, for instance, from a policy trained with Behavior Cloning (BC).
> >
> > We chose to frame our work specifically for **offline RL** for a crucial reason: the guaranteed diversity of behaviors. Datasets from sources like D4RL, which are often collected from partially trained or exploring agents, are known to contain a rich and varied mixture of behaviors (e.g., sub-optimal, expert, exploratory). This makes them the perfect testbed to validate a behavior *discovery* algorithm. In contrast, datasets from a single expert policy or standard BC often exhibit very little behavioral diversity, which would make it difficult to demonstrate our method's primary contribution.
> >
> > Therefore, while the framework is broadly applicable, our focus on RL is a deliberate methodological choice to test our approach on data where its strengths can be most rigorously and clearly demonstrated. We agree this broader applicability is a promising direction for future work.

---

> > > ### Author Response · Authors · 2025-07-22
> > > **Author response to Reviewer Ttp7**
> > >
> > > Dear Reviewer,
> > >
> > > We Thank you again for the efforts you put in to improve our work with your inputs.
> > >
> > > As the discussion phase is coming to an end, we kindly invite you to review our responses to your comments. If you have any further questions, please don’t hesitate to reach out. If no further concerns remain, we would greatly appreciate it if you updated your recommendation.
> > >
> > > Best Regards,
> > > Authors of Paper 5129

---

### Review · Reviewer_CtAc · 2025-07-05

**Summary Of Contributions:**

In this paper, the authors addresses the limitations of current explainability methods in reinforcement learning, which often overlook recurring strategies and temporally extended behavior patterns. The authors propose a fully offline, reward-free framework that segments agent behavior into interpretable clusters across trajectories, enabling fine-grained attribution of actions to coherent behavior segments. Unlike prior methods that focus on isolated states or entire episodes, this approach captures and explains recurring decision-making patterns. In experiment, four offline RL environments demonstrate that the proposed  method produces more faithful, human-preferred, and coherent explanations than existing SOTA baselines. Further, the reviewer thanks the authors for making their code publicly available.

**Audience:**

Yes

**Broader Impact Concerns:**

The impact is good, potentially has lots of applications in healthcare and other domains

**Claims And Evidence:**

Yes

**Requested Changes:**

Please see weaknesses

**Strengths And Weaknesses:**

Pros:

1.The research problem in interesting and important. While I am not an expert in RL, the introduction about the applications of explainability in Robotics, healthcare sounds interesting. With more RL algorithms being deployed in the real world, understanding them will play an increasingly important role.

2.The limitations are clearly discussed as a separate section. Potential future directions are also provided to address the limitations.

3.Most figures are well-designed, with good illustration that help the readers to understand the main contributions of this paper.



Cons:

1.Baseline models. It seems the baselines compared in the experiment section are quite limited. Could you explain the rationale for this? Also, why these methods are selected. The reviewer understands the specific challenges in terms of comparison and I am not asking for more experiments. More justifications will be appreciated.
At the beginning of section 4, please add references to these benchmark environments.

2.In figure 3, is the goal only for example demonstration? Or what messages this figure tries to deliver. While this figure is clear but seems lacks enough readable information.

3.There are 4 sub-components in figure 2. The reviewer suggests the authors to label them with 1234 or abcd, which probably will improve the readability when referred in the main paper.

4.The reviewer is also curious about how the explainable results will benefit specific applciaitons such as in healthcare. Could you provide some examples? Who will use the results and how to benefit from them.

---

> ### Author Response · Authors · 2025-07-12
> **Response to reviewer CtAc**
>
> We sincerely thank the reviewer for their positive feedback on the importance of the research problem, the clarity of our limitations section, and the design of our figures. We are encouraged by their thoughtful comments and offer the following clarifications.
>
> ### **Rationale for Baseline Selection**
>
> We appreciate the reviewer's question about our choice of baselines. Our primary comparison is with **Deshmukh et al. (2024)** because it is the most relevant and, to our knowledge, the only existing work that explains RL decisions by attributing them to **entire trajectories** from an offline dataset. This makes it the ideal baseline to highlight the core novelty of our approach: moving from coarse, trajectory-level explanations to fine-grained, **behavior-level** explanations.
>
> Our goal was to directly compare these two paradigms. While Deshmukh et al. conducted experiments on three environments, we validated our method on those same three **and** added the `pen-expert-v1` environment. This is a particularly challenging, high-dimensional robotic manipulation task, which helps demonstrate the scalability and effectiveness of our behavior-centric approach on a wider range of complex problems.
>
> ### **Purpose of Figure 3**
>
> The reviewer is correct that Figure 3 is for demonstration. Its main purpose is to provide a clear, visual example of our method applied to the **HalfCheetah** environment. The figure illustrates how a single, continuous trajectory is automatically partitioned into distinct, color-coded behavioral segments, such as hind-leg loading before a leap (Cluster 0) and high-speed jumping (Cluster 2). This figure serves the same illustrative role as Figure 1 (MiniGrid), Figure 4 (Pen), and Figure 5 (Seaquest), showing that our framework successfully segments behaviors across diverse environments.
>
> ### **Labeling in Figure 2**
>
> This is an excellent suggestion for improving readability. The four sub-components of our framework in Figure 2 are currently labeled with small numbers (1, 2, 3, 4). However, we agree that these could be missed. In our revision, we will make these labels more prominent to ensure the diagram is easier to reference from the main text.
>
> ### **Application in Healthcare**
>
> We thank the reviewer for asking for a concrete example in a critical domain like healthcare. Our method offers a significant advantage over prior work that attributes outcomes to whole trajectories.
>
> Consider a scenario in an ICU where patient data is collected offline. An RL agent recommends patient treatments. A patient is admitted, receives a standard, correct dose of antibiotics, but three hours later suffers a fatal bleeding event. A method that attributes this catastrophic outcome to the **entire trajectory** would be misleading; it would incorrectly associate the valid antibiotic treatment with the negative outcome.
>
> Our behavior-centric approach would resolve this ambiguity. It would segment the patient's trajectory into distinct behavioral clusters. For instance, it could identify "stable response to antibiotics" as one behavior and a subsequent, separate "hemorrhagic shock onset" as a different, critical behavior.
>
> * **Who benefits?** Clinicians and medical auditors.
> * **How do they benefit?** Instead of a confusing signal that the entire treatment plan was flawed, they would receive a precise explanation pointing to the specific physiological phase and behavior (the bleeding) that led to the failure. This allows them to diagnose the true problem and refine the agent's strategy for handling such specific events in the future, which is crucial in high-stakes, offline settings like healthcare.

---

> > ### Author Response · Authors · 2025-07-22
> > **Author response to Reviewer CtAc**
> >
> > Dear Reviewer,
> >
> > We Thank you again for the efforts you put in to improve our work with your inputs.
> >
> > As the discussion phase is coming to an end, we kindly invite you to review our responses to your comments. If you have any further questions, please don’t hesitate to reach out. If no further concerns remain, we would greatly appreciate it if you updated your recommendation.
> >
> > Best Regards,
> > Authors of Paper 5129

---

### Review · Reviewer_qhiW · 2025-07-06

**Summary Of Contributions:**

The authors propose a behaviour discovery algorithm for explainable reinforcement learning. The algorithm consists of two phases. In the first phase, it trains a transformer-based Vector Quantized Variational AutoEncoder (VQ-VAE) to predict the next state based on a sequence of states and actions. Then, it applies graph segmentation to cluster the behaviour segments. Extensive quantitative and qualitative studies were conducted on various offline RL benchmarks to demonstrate the efficacy of the proposed algorithm.

**Audience:**

Yes

**Broader Impact Concerns:**

I do not have any concerns on the ethical implications of the work.

**Claims And Evidence:**

Yes

**Requested Changes:**

All of the following are non-critical.
1. The paper does not cite the D4RL paper Fu et al. (2020).
2. Fu et al. (2020) states that the medium-level datasets were collected using partially trained SAC policies, not PPO policies.
3. The original D4RL benchmark does not contain the SeaQuest environment.

### Reference
Fu, Justin, et al. "D4rl: Datasets for deep data-driven reinforcement learning." arXiv preprint arXiv:2004.07219 (2020).

**Strengths And Weaknesses:**

The paper presents extensive quantitative and qualitative analysis on the efficacy of the proposed algorithm. Also, it is well-written and easy to understand. However, justification for using a state-prediction loss is missing. As the goal is to discover behaviours, wouldn't it be more natural to predict actions instead of states? Also, the probability distribution of the next state given the current state and action is the transition function, which is the characteristic of the MDP and is independent of the agent's behaviour. In fact, the encoder does not even need to encode the whole segment due to the Markovian property of the MDP. Only attending to the last state-action pair of the segment will suffice for predicting the next state. Yet, the experimental results seem to imply that the encoder is indeed attending to the entire segment and is encoding segments from different behaviours into different codes. Why does it work?

---

> ### Author Response · Authors · 2025-07-12
> **Response to reviewer qhiW**
>
> We sincerely thank the reviewer for their constructive feedback and insightful questions. We are glad they found our analysis extensive and the paper easy to follow. Below, we address the points raised.
> ### Justification for State-Prediction Loss
>
> We thank the reviewer for this important question. Our choice of a state-prediction objective is a deliberate design decision, motivated by its proven ability to learn rich behavioral representations from sequential data.
>
> * **Alignment with Baselines**: Our primary motivation stems from similar and successful approaches. Specifically, the most relevant baseline, Deshmukh et al. [1], states they train an "LSTM-based trajectory encoder following the procedure of trajectory transformer" and this is confirmed in their codebase as well [5]. The Trajectory Transformer framework is an autoregressive model that learns by predicting the next element in a sequence of states and actions. This means their encoder is fundamentally trained on a predictive objective that involves modeling future states from past states and actions, validating our general approach. Our work builds on this by using a more advanced transformer architecture and a different clustering and attribution pipeline.
>
> * **Learning Powerful Representations from Prediction**: Previous research on world models has shown that predicting future states is a powerful self-supervised objective for learning effective representations. By "effective," we mean representations that enable more sample-efficient learning and the derivation of high-performing policies, as foundationally demonstrated by works like Dreamer [2]. Our approach is inspired by this principle, and other prominent methods like Trajectory Transformer [3] also successfully utilize state prediction as part of their sequence modeling architecture to learn from offline data.
>
>
> * **On Using State Prediction vs. Action Prediction**: Our choice to use state prediction as the primary learning objective is a direct approach to modeling the consequences of an agent's actions. While predicting actions can also be effective, it is often used for a different purpose. For instance, some methods use an inverse dynamics model (`p(a|s,s')`) not as the final objective, but as a powerful self-supervisory task to learn a feature space that is robust to irrelevant environmental factors. A forward (state-prediction) model is then trained within this feature space to generate an intrinsic reward [4]. Our approach is a valid alternative, using a VQ-VAE trained directly on a state-prediction task to learn these behavioral representations.
>
>
> * **Modeling Behaviors as Subsequences**: Our core goal is to model *behaviors* as meaningful subsequences within a trajectory, just as our baseline attempts to do at the full trajectory level. For this purpose, using an action-conditional state prediction loss is a proven and sufficient method for learning representations that capture the necessary dynamics for subsequence clustering, as demonstrated by prior work [2, 3].
>
> * **Capturing Temporally-Extended and Robust Strategies**: The reviewer correctly notes that for a Markovian environment, only the last state-action pair is needed to predict the next state. However, our goal is not to model the environment's one-step dynamics, but to discover the structure of the agent's **policy**. Agent strategies or behaviors (e.g., "recovering from a fall," "approaching a goal") are often temporally extended patterns, even in Markovian settings. A behavior can be as short as one action (e.g., "picking up a key"), but many complex strategies unfold over time. Our transformer architecture is specifically designed to capture these longer-term dependencies. Furthermore, a key challenge is to learn representations that are invariant to nuisance aspects of the environment (e.g., moving leaves, changing textures) that do not affect the agent. While some methods use an inverse dynamics model to explicitly filter out such information [4], our approach achieves a similar robustness by discretizing behavior with a VQ-VAE and then identifying coherent, high-level clusters, effectively abstracting away irrelevant low-level details.
>
> ### Factual Changes
>
> We appreciate the careful reading and will make the following corrections:
>
> * **D4RL Citation**: We will ensure the existing citation for Fu et al. (2020) is clearly placed.
> * **PPO vs. SAC Policy**: Corrected; the `halfcheetah-medium-v2` dataset used SAC policies.
> * **SeaQuest Environment**: Corrected; we will clarify this dataset is not from the original D4RL benchmark.
>
> We hope these clarifications are helpful and thank the reviewer again for their feedback.
>
> ### References
> [1] Deshmukh et al., ICLR (2023).
> [2] Hafner et al., ICLR (2020).
> [3] Janner et al., NeurIPS (2021).
> [4] Pathak et al., ICML (2017).
> [5] https://github.com/shripaddeshmukh/xrl_with_trajectories/tree/main

---

> ### Author Response · Authors · 2025-07-22
> **Author response to reviewer qhiW**
>
> Dear Reviewer,
>
> We Thank you again for the efforts you put in to improve our work.
>
> As the discussion phase is coming to an end, we kindly invite you to review our responses to your comments. If you have any further questions, please don’t hesitate to reach out. If no further concerns remain, we would greatly appreciate it if you updated your recommendation.
>
> Best Regards,
> Authors of Paper 5129

---

### Author Response · Authors · 2025-07-12
**Summary of the changes**

### **Summary of Revisions**

We would like to extend our sincere thanks to the reviewers for their thorough and constructive feedback, which has been invaluable in improving our work. We have carefully considered all comments and have revised the manuscript to address the points raised. For ease of review, all significant changes in the text have been highlighted in blue.

The main changes can be summarized as follows:

* **Methodological Clarifications:** In our response to Reviewer qhiW, we provide a detailed justification for our model's state-prediction objective. We clarify its relationship to baseline methods like Deshmukh et al. and its effectiveness in learning robust behavioral representations by citing established work in world models and self-supervision. We further explain why our sequence-based approach is necessary for discovering the temporally-extended structures present in an agent's policy.

* **Strengthened Experimental Validation:** In our response to Reviewer **Ttp7**, we address the concerns about our experimental evaluation by emphasizing the strong **quantitative results** (e.g., Average Fidelity Score) that support our qualitative findings. We also provide a detailed clarification of our human study design, explaining its robustness and how it aligns with standard practices in XAI evaluation to demonstrate the validity of our approach.

* **Improved Presentation and Context:** Based on feedback from all reviewers, we have improved the paper's overall presentation. This includes:
    * Enhancing the visualizations in all figures with clearer, color-coded backdrops.
    * Revising figure captions for improved clarity and precision.
    * Adding a concrete application example in healthcare to illustrate the method's real-world utility (Reviewer **CtAc**).
    * Correcting all factual points raised regarding the D4RL datasets and other minor details.

We hope these changes satisfactorily address the reviewers' concerns. We thank you again for your time and valuable input.

---

### Decision · Action_Editor_M89P · 2025-08-11

**Recommendation:** Accept as is

**Audience:**

Yes

**Audience Explanation:**

The reviewers and I agree that the community interested in understanding the behavior of RL agents (or as a reviewer pointed out, other sequential decision making systems) could be interested in using this approach as a part of such research.

**Claims And Evidence:**

Yes

**Claims Explanation:**

The reviewers and I agree that the experiments show some benefit of the method as a way of identifying more semantically meaningful components of behavior, as demonstrated across multiple experimental domains.